# Phase I Study of a Combination of Fluvastatin and Celecoxib in Children with Relapsing/Refractory Low-Grade or High-Grade Glioma (FLUVABREX)

**DOI:** 10.3390/cancers15072020

**Published:** 2023-03-28

**Authors:** Pierre Leblond, Emmanuelle Tresch-Bruneel, Alicia Probst, Nadège Néant, Caroline Solas, Arthur Sterin, Thomas Boulanger, Isabelle Aerts, Cécile Faure-Conter, Anne-Isabelle Bertozzi, Pascal Chastagner, Natacha Entz-Werlé, Emilie De Carli, Marie-Cécile Le Deley, Gauthier Bouche, Nicolas André

**Affiliations:** 1Pediatric Oncology Unit, Oscar Lambret Comprehensive Cancer Center, 59037 Lille, France; pierre.leblond@ihope.fr; 2Centre Léon Bérard, Institut d’Hématologie et d’Oncologie Pédiatrique (IHOPe), 69008 Lyon, France; cecile.conter@ihope.fr; 3Department of Clinical Research and Innovation, Oscar Lambret Comprehensive Cancer Center, 59037 Lille, France; e-tresch@o-lambret.fr (E.T.-B.); a-probst@o-lambret.fr (A.P.); 4Laboratoire de Pharmacocinétique et Toxicologie, La Timone, AP-HM, 13005 Marseille, France; nadege.neant@ap-hm.fr (N.N.); caroline.solas@ap-hm.fr (C.S.); 5Service d’Hématologie et Oncologie Pédiatrique, La Timone, AP-HM, 13005 Marseille, France; arthur9200@gmail.com; 6Imaging Department, Oscar Lambret Comprehensive Cancer Center, 59037 Lille, France; t-boulanger@o-lambret.fr; 7Institut Curie, PSL Research University-Oncology Center SIREDO, 75248 Paris, France; isabelle.aerts@curie.fr; 8Pôle de Pédiatrie Unité Hemato-Immuno-Oncologie, Toulouse University Hospital, 31300 Toulouse, France; bertozzi.ai@chu-toulouse.fr; 9Service d’Hémato-Oncologie Pédiatrique, Nancy University Hospital, 54500 Vandoeuvre-lès-Nancy, France; p.chastagner@chru-nancy.fr; 10Pédiatrie Onco-Hématologie—Pédiatrie III, Strasbourg University Hospital, 67091 Strasbourg, France; natacha.entz-werle@chru-strasbourg.fr; 11Service d’Hémato-Oncologie Pédiatrique, Angers University Hospital, 49100 Angers, France; emdecarli@chu-angers.fr; 12Department of Biostatistics, Oscar Lambret Comprehensive Center, 59037 Lille, France; m-ledeley@o-lambret.fr; 13The Anticancer Fund, 1860 Meise, Belgium; gauthier.bouche@anticancerfund.org; 14CRCM INSERM U1068 SMARTc, Aix Marseille University, 13007 Marseille, France; 15Metronomics Global Health Initiative, 13005 Marseille, France

**Keywords:** high-grade pediatric glioma, low-grade pediatric glioma, phase I trial, drug repurposing

## Abstract

**Simple Summary:**

This study tested the repurposing of two rationally selected, non-anticancer drugs as a way to address the need for less toxic therapeutic options in children with gliomas. The determined recommended phase II dose of fluvastatin in combination with celecoxib in children with gliomas is 6 mg/kg/day (in 14 days on, 14 days off schedule) with a fixed daily dose of celecoxib (from 200 mg to 800 mg depending on weight). The combination is not active in HGG but could be explored as a maintenance treatment in LGG patients to avoid or delay a possible tumor recurrence, which would require a more toxic treatment. This oral strategy with very limited toxicity may be used to gain time and therefore limit treatment-related toxicities in growing children. Given its good safety profile, its low cost and all-oral administration, we think that it could be considered as an option for children with LGG living in low- and middle-income countries.

**Abstract:**

Preclinical data support the activity of celecoxib and fluvastatin in high-grade (HGG) and low-grade gliomas (LGG). A phase I trial (NCT02115074) was designed to evaluate the safety of this combination in children with refractory/relapsed HGG and LGG using four dose levels of fluvastatin with a fixed daily dose of celecoxib. A Continual Reassessment Method was used for fluvastatin dose escalation. Dose-limiting toxicities (DLT) were determined on the first treatment cycle. Twenty patients were included. Ten LGG and ten HGG patients received a median of 3.5 treatment cycles. Two DLTs were reported: one grade 3 maculopapular rash (4 mg/kg dose level) and one grade 4 increase of Creatine Phospho-Kinase (6 mg/kg dose level). We identified the dose of 6 mg/kg/day as the recommended phase II dose (RP2D) of fluvastatin with celecoxib. Four patients with LGG continued treatment beyond 12 cycles because of stable disease, including one patient who received 23 treatment cycles. In children with refractory/relapsed glioma, the RP2D of fluvastatin with celecoxib is 6 mg/kg/day. The long-term stable diseases observed in LGG suggest a possible role of the combination in a maintenance setting, given its good tolerance and low cost for children living in low- and middle-income countries.

## 1. Introduction

Gliomas are the most common pediatric central nervous system (CNS) tumors [1]. Overall survival (OS) at 10 years ranges between 85% and 96% for low-grade gliomas (LGG), whereas nearly all children with high-grade gliomas (HGG) die within 1–2 years of diagnosis [1,2,3,4,5]. The therapeutic challenges dramatically differ, with the management of HGG critically in need of new thinking [6]. For LGG, the main objective is to obtain long-term tumor control while limiting treatment-induced long-term effects. While surgery is very effective, resectability is particularly challenging for optic pathway gliomas (OPG) [7]. Radiotherapy effectively controls tumor growth in most cases but carries a high risk of inducing long-term sequelae [8,9,10,11,12]. Chemotherapy has increasingly become the mainstay of pediatric LGG management. Commonly used regimens include carboplatin and vincristine, vinblastine, thioguanine, procarbazine, lomustine and vincristine (TPCV) and, more recently, bevacizumab [13,14,15,16]. Chemotherapy is however frequently only transiently effective, and multiple lines of chemotherapy are often required [13]. Therefore, there is a need for new combinations using drugs with favorable short- and long-term safety profiles. 

In a preclinical study comparing the transcriptional profiles of five hypothalamo–chiasmatic and six cerebellar pilocytic astrocytomas using a microarray technique, quantitative real-time PCR and immunochemistry, Tchoghandjian et al. demonstrated that these entities are genetically distinct by showing many differentially upregulated genes [17]. These results were confirmed by Mercurio et al., who proposed that celecoxib and fluvastatin could target a set of these genes (ICAM1, CRK, CD36 and IQGAP1) differentially expressed in OPG [18]. Fluvastatin is an approved HMG-CoA reductase inhibitor that has been tested in two pediatric cancer trials with no safety concerns and encouraging efficacy [19,20]. Of relevance to glioma, statins target CD36, a scavenger receptor that is highly expressed in pilocytic astrocytoma [21]. Fluvastatin also showed an inhibitory and cytotoxic effect on several high-grade glioma cell lines [18,22,23]. Celecoxib is a cyclooxygenase 2 (COX-2) inhibitor and has a long repurposing history in oncology [24]. Sato et al. showed that the expression level of COX-2 was greater in LGG than in normal brain cells and that inhibiting COX-2 induced apoptosis and inhibited cell proliferation via the Akt/survivin and Akt/ID3 pathways in LGG [25]. Moreover, celecoxib interferes with cellular adhesion machinery by decreasing ICAM-1 expression and promotes anoikis by deregulating the focal adhesion assembly protein CRK-associated substrate, P130CAS [18]. Celecoxib has been used in several cancer trials [24] including in pediatric LGG protocols [26]. Mercurio et al. demonstrated a synergistic effect of fluvastatin and celecoxib in two glioma cell lines and reported a significant radiological response to this combination in a patient with refractory metastatic LGG [18]. 

Taking advantage of the safety and rationale to use these drugs against LGG, we conducted a phase I trial to determine the recommended phase II dose (RP2D) of fluvastatin in combination with a fixed dose of celecoxib in children with recurrent/refractory glioma. Considering the low toxicity of each of these two drugs and their distinct mechanism of action, no significant toxicity was expected from their combination. The trial included an expansion cohort of LGG patients to better characterize the drugs’ safety and assess the preliminary activity of the combination at the recommended phase II dose (RP2D).

## 2. Materials and Methods

### 2.1. Study Population

Patients aged 6–21 years were eligible if they met all the following criteria: refractory/recurrent LGG or HGG after at least one first-line therapy, including radiotherapy for HGG patients; Lansky play scale or Karnofsky performance status at least 70%; ability to swallow oral medications; life expectancy of at least 3 months; adequate bone marrow (absolute neutrophil count ≥1000/μL, platelet count ≥75,000/μL), renal (1.5 × age-adjusted normal serum creatinine or glomerular filtration rate ≥70 mL/min/1.73 m^2^ with the Schwartz formula), liver (total bilirubin ≤ 3 N and alanine aminotransferase ≤ 4 N) and muscle enzyme (with Creatine Phospho-Kinase CPK < 2 N) functions; no cytotoxic chemotherapy within 21 days (2 weeks if vincristine and 6 weeks if nitrosourea) and no radiotherapy within 6 months prior to study entry.

Prior histological documentation was not required in patients with Neurofibromatosis-1 (NF1) and typical radiologic low-grade OPG. Disease refractory was defined as a radiographic or clinically progressive disease while on treatment. Patients with LGG had to be considered non-eligible for complete tumor resection. Patients with HGG had to have histologically confirmed recurrent or progressive disease. Patients with completely resected HGG at relapse were eligible. Patients with LGG must have had measurable lesions according to the RANO criteria [27]. Patients with diffuse intrinsic pontine gliomas were not eligible. 

Exclusion criteria were active uncontrolled infection, peptic ulcer disease, gastrointestinal bleeding, inflammatory bowel disease, history of asthma, acute rhinitis or nasal polyps, medical history of allergy or hypersensitivity induced by acetylsalicylic acid or non-steroidal anti-inflammatory drugs or known hypersensitivity to sulfonamides, non-congestive heart failure, ischemic heart disease, cardiovascular disease, preexisting muscle pathology, pregnancy, breastfeeding and organ toxicity superior to grade 2 according to NCI-CTCAE v4.0.

Patients and/or their legal guardians gave written informed consent, and assent was obtained as appropriate at the time of enrollment. The protocol and amendments received regulatory approvals from independent ethics committees and complied with the French regulations and the Declaration of Helsinki.

### 2.2. Study Design and Treatment

FLUVABREX (NCT02115074) was a national, multicentric, interventional, open-label, non-comparative and non-randomized dose-escalation study. The main objective was to determine a recommended phase II dose (RP2D) of fluvastatin when combined with celecoxib based on a dose-limiting toxicity (DLT) evaluation (primary endpoint). The secondary objectives were to assess the safety of the drug combination and pharmacokinetics (PK) of both drugs to assess the progression-free survival (PFS) and the overall survival (OS) and to describe the best tumor response according to the RANO criteria (secondary endpoints). Ten early-phase pediatric oncology centers from the French National Pediatric Oncology Society (SFCE) participated in this study.

Fluvastatin was orally given once daily from day 1 to day 14 of the 28-day cycle. Four dose levels, defined according to a previous study, were planned: 2 mg/kg/day (level 1), 4 mg/kg/day (level 2), 6 mg/kg/day (level 3) and 8 mg/kg/day (level 4) [19]. Celecoxib was orally given daily from day 1 to day 28 of each 28-day cycle at a fixed dose according to weight: 100 mg twice daily (BID) if <20 kg, 200 mg BID if 20–50 kg and 400 mg BID if >50 kg.

### 2.3. Dose-Escalation Phase

The dose-escalation scheme only concerned fluvastatin and was based on a Continual Reassessment Method Likelihood approach (CRML) model [28]. The dose associated with a probability of DLT closest to 25% was considered the recommended dose. The first patients were included at level 1 (2 mg/kg/day of Fluvastatin). Patients were included in cohorts of a minimum of 2 patients. For each dose level, the second patient could only be included after the first patient was evaluated. Escalation by a cohort of 2 patients was continued until the first DLT was observed. As soon as the first DLT was observed, the CRML model was activated to estimate a posteriori probabilities of toxicity associated with each dose level. The next patient was treated at the dose level corresponding to the estimated probability of DLT closest to the target, i.e., 25%, without skipping a level. The a priori probabilities of toxicities were as follows: 0.04, 0.07, 0.20, 0.33 and 0.50 for 1, 2, 4, 6 and 8 mg/kg/day of fluvastatin. If a patient stopped the treatment during the first cycle or received treatment at a reduced dose (<75% of the expected dose according to the protocol) for a reason other than toxicity, they were considered non-evaluable and was replaced.

The total number of patients depended on the dose level identified as the recommended dose as well as the number of DLTs observed at each dose level. The recommended dose for a possible phase II trial (RP2D) was defined as the dose level with a probability of DLT below 33% and the closest to 25%. The escalation phase ended when 6 evaluable patients had been included at the recommended dose.

### 2.4. Expansion Phase

Following completion of the dose escalation part, the protocol was amended to include patients with LGG only, and the number of subjects was increased to reach a total of 14 evaluable patients treated at the current recommended dose or upper dose in order to better characterize the safety and assess preliminary efficacy in a sufficient number of LGG patients. Patients of the expansion phase were included in the CRML model to confirm or not the recommended dose, allowing for subsequent dose re-escalation. If the dose was increased during the expansion phase, a minimum of 6 patients had to be treated and evaluated at the revised RP2D before stopping the trial.

During the dose-escalation and the expansion phases, intra-patient dose escalation was not allowed. Treatment was administered until progression or unacceptable toxicity for one year. Treatment continuation beyond one year was possible in case of clinical and/or radiological benefit (response or stabilization), depending on the patient and investigator’s decision.

### 2.5. Safety Evaluation

The safety of the study treatment was evaluated based on clinical and biological evaluations, including a complete blood count, biochemistry tests, liver and kidney functions and CPK levels at day 1 and day 14 during the first cycle, then at day 1 of each cycle. Toxicities were evaluated according to the NCI-CTC v4.0 criteria.

DLTs were evaluated during the first cycle (28 days) and were defined as follows: grade 3 or 4 neutropenia leading to temporary treatment discontinuation for more than 7 days, grade 3 or 4 thrombocytopenia requiring transfusions for more than 7 days or grade 3 or 4 non-hematologic toxicities. Exceptions were the following events: nausea and vomiting, grade 3 fever and grade 3 liver rapidly recovering toxicity, and grade 3 increase of CPK levels but rapidly reversible (back to <3 × normal within 2 weeks after treatment interruption). Toxicity leading to significant dose reduction (<75% dose as per protocol) was also considered as a DLT, even if the grade of toxicity did not in itself justify this classification. All adverse events were reported over the whole treatment duration except those related to the underlying disease or its progression. 

### 2.6. Pharmacokinetic Assessments

Blood pharmacokinetic (PK) samples were collected on days 1 (D1) and 14 (D14) of cycle 1 at pre-dose, 0.5, 1, 2, 3, 4, 5, 8, 12 and 24 h post-dose of fluvastatin and 12 h after the second dose of celecoxib. Celecoxib plasma trough concentration (Ctrough) was collected on day 28 (D28) of cycle 1. Fluvastatin and celecoxib plasma concentrations were determined using a validated sensitive ultra-performance liquid chromatography coupled to tandem mass spectrometry (LC-MS/MS) method with a lower limit of quantification of 0.1 ng/mL and 10 ng/mL, respectively. Precision and accuracy were within the ±15% over the calibration range (fluvastatin from 0.1 ng/mL to 100 ng/mL and celecoxib from 10 ng/mL to 2000 ng/mL).

PK parameters of both fluvastatin and celecoxib were estimated by standard non-compartmental analysis using Monolix 2019 software (Lixoft, Orsay, France). The maximum observed plasma concentration (Cmax) and the time to maximum observed plasma concentration (Tmax) were directly determined from the plasma concentration–time profile for each patient. The area under the concentration–time curve from time zero to the last measurable concentration (AUC0-tlast), the area under the concentration–time curve extrapolated to infinity (AUC_0–∞_), the terminal elimination phase half-life (t1/2), the apparent oral clearance (CL/F) and the apparent volume of distribution during the terminal phase (Vz/F) were assessed. CL/F and Vz/F were normalized for body weight.

### 2.7. Efficacy Evaluation

Best tumor response was defined according to the RANO criteria in patients who received at least 2 cycles of treatment [27]. It was separately determined for all evaluable patients and for all patients treated at the RP2D. Brain MRI and spinal MRI, if needed, with at least 2 plans of gadolinium-enhanced T1 sequences (sagittal, axial and/or coronal), with T2 and FLAIR sequences, were performed every 3 cycles. Imaging was centrally reviewed. PFS was defined as the time from study entry to the date of progression or death, whichever occurred first. OS was defined as the time from study entry to the date of death of any cause. In the absence of any event, patients were censored at the date of the last follow-up. The distribution of follow-up was estimated using the reverse Kaplan–Meier method.

## 3. Results

### 3.1. Patient and Tumor Characteristics

From June 2014 to October 2018, 20 patients were enrolled (cf. Figure 1 and Figure 2), including 13 patients in the dose-escalation phase and 7 in the expansion phase. Characteristics of the patients are summarized in Table 1. 

The median age of patients at inclusion is 12.5 years, with a median time from the initial diagnosis to registration in the study of 29 months (range: 3–173). Ten patients (50%) had an HGG, and ten patients (50%) had an LGG, including three NF1 patients. At inclusion, all patients had already received at least one line of chemotherapy or targeted therapy (median: 3, range 1–7). Surgery had been performed in 16 (80%) patients, and 12 (60%) patients had received prior radiation therapy.

Tumor and treatment characteristics of LGG and HGG patients are summarized in Table 2.

### 3.2. Treatment Exposure

During the dose-escalation phase, five patients were treated at dose level 1 of fluvastatin, seven patients at level 2 and one patient at level 3 (Figure 1 and Figure 2). During the expansion phase, two patients were treated at level 2 and five patients at level 3. The median number of cycles is 3.5 (range, 1–23) in the overall population, 3 (1–4) for HGG patients and 9 (1–23) for LGG patients. Among patients with LGG, three patients received 3 cycles or less, three patients received 6 to 9 cycles and four patients received 12 or more cycles (12, 18, 18 and 23 cycles). 

Sixteen patients (80%) received a complete first course of treatment according to the protocol. Three (15%) patients were not evaluable for DLTs for the following reasons: lower dose received per investigator’s decision because of obesity, tumor progression during the first cycle (one patient), vomiting starting on day 4 (unrelated to study treatment) resulting in an inability to swallow the drug and leading to permanent study treatment discontinuation (one patient). One (5%) patient presented with a treatment-related, grade 3 maculopapular rash (DLT), leading to permanent treatment discontinuation after 22 days of treatment.

Most patients stopped treatment for progressive disease (*n* = 14, 70%), including two LGG patients who, according to the local radiological assessment, had progressive disease. Of note, both of them were later not considered as a progression by the central review, according to the RANO criteria. One patient (5%) stopped for DLT, followed by an early disease progression, and one other (5%)—as previously mentioned—stopped after 4 days of treatment because of vomiting unrelated to the study treatment. Four (20%) patients completed treatment according to the protocol with 12 cycles or more.

### 3.3. Toxicities

Seventeen patients (85%) were evaluable for the DLTs (Table 3). DLTs were observed in two patients during the dose-escalation phase. One patient treated at level 2 (4 mg/kg/d) presented with a grade 3 cutaneous rash after 17 days, resulting in definitive treatment discontinuation. The second patient presented with a DLT at level 3 (6 mg/kg/d) with a grade 4 increase in CPK after 13 days of treatment. The patient continued the treatment at level 2 with a reduced dose of fluvastatin for a total of 18 cycles. At the end of the escalation dose, level 2 and level 3 were associated with a probability of DLT of 20.6% (95% CI: 2.6–50.5%) and 33.6% (95% CI: 8.1–62.4%), respectively.

In the expansion phase, the dose was finally re-escalated to dose level 3 after two patients were treated at dose level 2 with no DLT, in accordance with the study’s protocol. The last five patients were treated at dose level 3 with no DLT. At the end of the study, the probability of DLT was 9.5% (95% CI: 1.3–28.0%) at level 2, 19.8% (95% CI: 5.0–41.6%) at level 3 and 36.3% (95% CI: 15.4–57.8%) at level 4. The RP2D was therefore determined to be 6 mg/kg/day (level 3).

All patients were evaluable for toxicity. Table 4 summarizes all grades ≥ 1 toxicity related to the fluvastatin–celecoxib combination observed during the whole study period. Of note, one patient presented with perturbation of hepatic function with transaminase and bilirubin increase (grade 3). The perturbation was transient (it had recovered at the subsequent evaluation, performed 16 days later) and was not considered a DLT.

The RP2D of fluvastatin in combination with celecoxib is 6 mg/kg/d given from day 1 to day 14 in 28-day cycles. Celecoxib is given daily from day 1 to day 28 at a fixed, body-weight-adjusted dose (see Methods).

### 3.4. Pharmacokinetic Analysis

Fluvastatin PK data were available for 13 and 10 patients on days 1 and 14, respectively. A summary of the PK parameters at each dose level and sampling day is reported in Table 5. An important inter-individual variability was reported, but there was no evidence of drug accumulation between D1 and D14, with a ratio ranging from 0.9 to 1.8, except for two patients at 4 mg/kg (accumulation ratio of 3.2 and 3.6). Celecoxib PK parameters from day 1 were obtained for 12 patients (Table 5). PK parameters at day 14 could not be calculated for all patients due to missing samples. Celecoxib Ctrough at day 28 of treatment was available in seven patients with a mean value (CV%) of 347 ng/mL (56.5%).

### 3.5. Efficacy

A total of 18 patients were evaluable for best response. Considering the nine evaluable LGG patients, the best response was a stable disease for eight patients (89%) and progression for one patient (11%). A minor radiological response was actually seen in one patient (Figure 3). 

All nine evaluable patients with HGG glioma had an early disease progression. All patients were evaluable for survival analysis. The median follow-up of living patients at the latest news was 40 months (range, 16.8–83.7 months). Twelve patients (ten HGG and two LGG) died after having experienced a disease progression (four at dose level 1, seven at dose level 2 and one at dose level 3). Median OS was 13.5 months (95% CI 3.7—not achieved) in the entire study population, 7.4 months (95% CI, 1.5–13.5 months) for HGG patients and not reached for LGG patients (Figure 4A). Median PFS was 2.8 months (95% CI 1.7–7.4 months) for the entire study population, 2.1 months (95% CI 0.95–2.6) for HGG patients and not achieved for LGG patients. Among patients with LGG, four patients are alive and free of disease progression at 17, 35, 41 and 84 months after study entry. The two patients who discontinued treatment because of radiological progression not confirmed by a central review were censored at the time of their last tumor evaluation in the study at 6 and 9 months, respectively (Figure 4B).

## 4. Discussion

The overall therapeutic management of patients with LGG exposes them to possible transient toxicities but also to long-term sequelae [8,9,10,11,12]. New treatment modalities with MEK inhibitors, BRAF inhibitors and metronomic scheduling could lead to long-term treatment with control of the disease [26,29,30]. In this trial, we investigated an oral drug repurposing strategy to reach this goal [31]. Treatment of HGG requires new agents and new thinking because of poor prognosis [6,32]. In this phase I study, the RP2D of fluvastatin in combination with celecoxib is 6 mg/kg/d given from day 1 to day 14 in 28-day cycles with continuous fixed doses of celecoxib. We observed two DLTs: one grade 3 maculopapular rash and one grade 4 elevation of CPK. In patients with LGG, seven of ten patients received the treatment for 6 cycles or more with stable disease. No partial or complete responses were observed. No activity was seen in HGG.

The experimental treatment displayed very good tolerability. Only two patients experienced DLT during the first cycle of treatment. One patient had a maculopapular rash grade 3 at the dose of 4 mg/kg/day but continued the treatment without toxicity at the dose of 2 mg/kg/day for 17 months. The second patient had a grade 4 increase in CPK at the dose of 6 mg/kg/day and stopped treatment. No patients stopped treatment because of treatment toxicities after the first cycle. More importantly, seven out of the ten LGG patients safely received at least 6 treatment cycles. The tolerability of the experimental treatment compares favorably to other therapeutic options [13,14,15]. Ater et al. reported peripheral nervous system grade 3–4 toxicity for 19% of patients treated with vincristine [15]. In a European randomized study, 84% of patients experienced grade 3–4 hematologic toxicity, 10% had an allergic reaction to carboplatin and 24% had grade 3–4 infections [13]. No patients treated with fluvastatin/celecoxib experienced hematologic toxicity or neurotoxicity. Verschuur et al. used celecoxib in a multidrug metronomic regimen in LGG patients and reported grade 3–4 neutropenia in 11 of 18 patients [26]. BRAF inhibitors offer a better safety profile, with the most frequent grade 3–4 adverse events being elevated Creatine Phospho-Kinase (10%) and maculopapular rash (10%) [30]. Similarly, the most frequent toxicities of MEK inhibitors were minor to moderately severe skin rash and gastrointestinal symptoms [29]. Here, we confirm the good safety profile of fluvastatin that López-Aguilar et al. reported in their pediatric cancer trial, with only low-grade gastrointestinal toxicity and myalgia [19]. 

The high inter-individual variability of fluvastatin we observed is consistent with prior PK findings [33]. Of note, no PK data are available for fluvastatin in the pediatric population, but PK parameters were consistent with those reported in the adult population except for the half-life, which was two-fold longer in our study (2.3 ± 0.9 h in adults) [34]. 

In terms of efficacy in LGG, 6-month PFS was 70% in this heavily pre-treated population. We did not observe any objective response, though seven of the ten patients received treatment for ≥6 cycles, and four patients were alive and free of progression with long follow-ups. Several therapeutic approaches have greater response rates and PFS than in our study. Gururangan et al. reported 3/30 (10%) partial responses (PR) and a 2-year PFS of 49% in patients treated with temozolomide [35]. In a metronomic phase II trial, Verschuur et al. reported two PR and six stable diseases (SD) in 10 LGG patients, including patients who had relapsed after or progressed on vinblastine. Additionally, 2-year PFS was 70%, and seven patients continued treatment beyond one year [26]. Bouffet et al. reported one CR and ten PR in 50 patients (22%) receiving weekly vinblastine with a 2-year PFS of 62% [14]. A recent PBTC study reported 2/35 (6%) PR and a 2-year PFS of 48% (±9%) for bevacizumab with irinotecan (B + I), but most patients relapse within 5 months after treatment cessation [36,37,38]. Roux et al. treated 16 patients with B + I, followed by metronomic maintenance with weekly vinblastine. After a median follow-up of 3.9 years after B + I cessation, nine of the sixteen patients were progression-free [39,40]. Because of the SD observed in our LGG patients, we think that the association of celecoxib and fluvastatin may represent an interesting maintenance approach for a patient treated with bevacizumab as those patients rapidly progressed after stopping treatment [37,38]. The potential place of this combination in the context of targeted therapies for LGG could also be explored. Aberrations of the MAPK pathway are key to oncogenesis in low-grade gliomas [41]. Sustained responses (36–40%) have now been observed in early-phase trials with MEK inhibitors, even in patients with multiple recurrences [29,30,42]. In our study, three patients presented with a KIAA1549-BRAF gene fusion. Interestingly, an 8-year-old male LGG patient had progressed on a MEK inhibitor before joining our trial. The patient experienced stable disease for 23 months while on treatment. A minor radiological response is actually seen in this patient (Figure 1). Modulation of autophagy might be an explanation of the observed response to the combination as was observed for chloroquine, though no strong biological data have been reported to date to support this hypothesis for fluvastatin and celecoxib [43,44].

## 5. Conclusions

In conclusion, this combination of fluvastatin with celecoxib displayed a good safety profile with interesting preliminary activity in children with LGG. Its use as a maintenance treatment may be worthy of further investigation in children with LGG. Of note, both drug repositioning and metronomic chemotherapy have been proposed as interesting therapeutic options for patients with cancer living in low- and middle-income countries (LMIC) as these treatments are orally available, are inexpensive, and display only limited toxicities [45]. Recently an international survey performed among pediatric oncologist working in LMIC has confirmed a growing interest in these strategies, which therefore represents a unique opportunity and shall be evaluated properly in this specific setting [46,47].

## Figures and Tables

**Figure 1 cancers-15-02020-f001:**
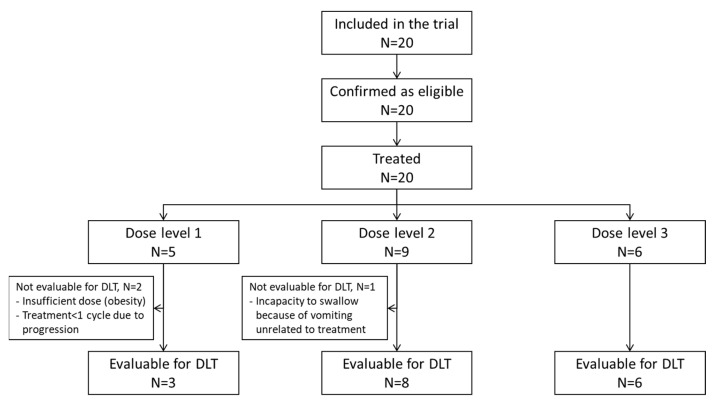
CONSORT diagram.

**Figure 2 cancers-15-02020-f002:**
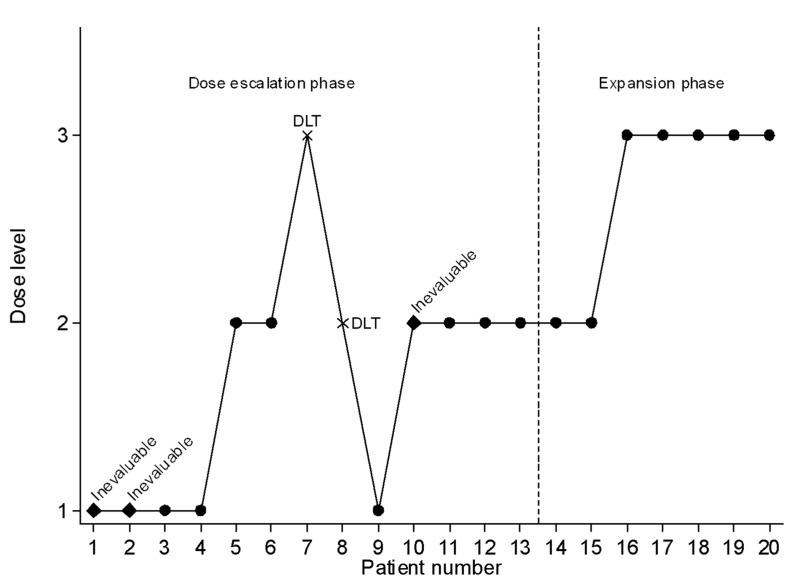
Sequential inclusion of patients with the allocated dose level.

**Figure 3 cancers-15-02020-f003:**
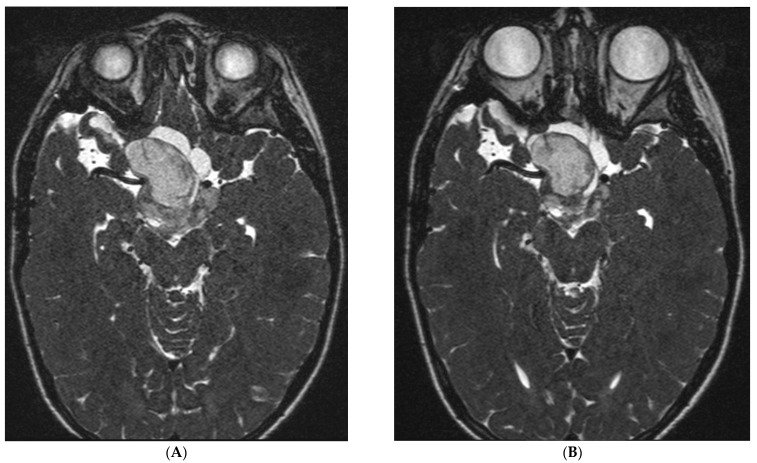
MRI (T2 gadolinium) of an LGG patient at inclusion (**A**) and 15 weeks later after 4 cycles of fluvastatin and celecoxib (**B**). This 8-year-old male patient experienced stable disease and remained on treatment for 23 cycles. A minor response, not meeting the RANO criteria, was observed when comparing the MRI at inclusion (**A**) with the MRI at week 15 (**B**).

**Figure 4 cancers-15-02020-f004:**
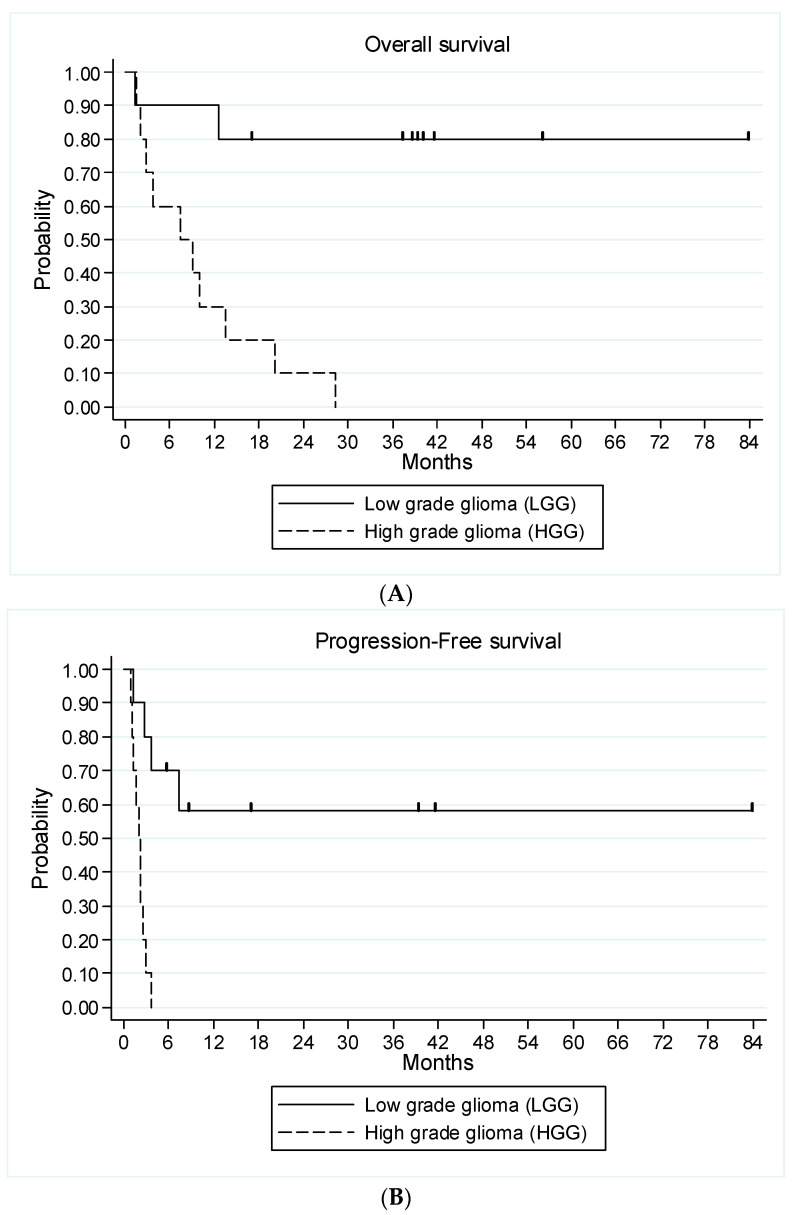
Kaplan–Meier curves for overall survival (**A**) and progression-free survival (**B**) of all 20 patients according to the type of glioma.

**Table 1 cancers-15-02020-t001:** Characteristics of the 20 patients.

Characteristics	N (%)
Sex	
Male	13 (65%)
Female	7 (35%)
Age in years	
Median (range)	12.5 (5.9–19)
Neurofibromatosis type 1	3 (15%)
Metastatic at study entry	2 (10%)
Grade	
Low-grade glioma	10 (50%)
High-grade glioma	10 (50%)

**Table 2 cancers-15-02020-t002:** Tumor and treatment characteristics per tumor grade (LGG and HGG).

	LGG (*n* = 10)		HGG (*n* = 10)	
Histology	Pilocytic astrocytoma	6 (30%)	Glioblastoma	6 (30%)
	Pilomyxoid astrocytoma	1 (5%)	Anaplastic glioma	2 (10%)
	Ganglioglioma (grade 2)	1 (5%)	Anaplastic oligodendroglioma	1 (5%)
	Diffuse astrocytoma (grade 2)	1 (5%)	Anaplastic oligoastrocytoma	1 (5%)
	No histology performed	1 (5%)		
BRAF rearrangement	Yes	3 (30%)		
	No	2 (20%)		
	Unknown	5 (50%)		
BRAF V600E mutation	Yes	0 (0%)		
	No	8 (80%)		
	Unknown	2 (20%)		
Prior treatment	Targeted or chemo-therapy	10 (100%)	Chemotherapy	10 (100%)
	Radiation therapy	2 (20%)	Radiation therapy	10 (100%)
	Surgery	7 (70%)	Surgery	9 (90%)
No. of prior lines	Median (range)	4 (2–7)	Median (range)	1.5 (1–4)
Previous therapy	Vinblastine	10 (100%)	Temozolomide	10 (100%)
	Vincristine–Carboplatin	9 (90%)	Bevacizumab	5 (50%)
	Bevacizumab–Irinotecan	3 (30%)	Lomustine	4 (40%)
	Bevacizumab alone	3 (30%)	Other regimens ^b^	7 (70%)
	Thioguanine–procarbazine–lomustine–vincristine (TPCV)	3 (30%)		
	Other regimens ^a^	8 (80%)		

^a^ BBSFOP regimen (procarbazine, carboplatin, vincristine, cisplatin, etoposide and cyclophosphamide) *n* = 2; SFCEMetro01 regimen (vinblastine, cyclophosphamide, methotrexate and celecoxib) *n* = 1; temozolomide *n* = 1; vinorelbine *n* = 1, cobimetinib *n* = 1; cyclophosphamide *n* = 1 and cisplatin *n* = 1. ^b^ Oral etoposide, topotecan, regorafenib, hydroxyurea, carboplatin, adavosertib and pomalidomide, *n* = 1 for each drug.

**Table 3 cancers-15-02020-t003:** Number of patients per fluvastatin dose levels and DLTs.

Level (Dose)	Number of Patients	Evaluable for DLTs	DLT—Number (Details)
1 (2 mg/kg/d)	5	3	/
2 (4 mg/kg/d)	9	8	1 (grade 3 maculopapular rash)
3 (6 mg/kg/d)	6	6	1 (grade 4 CPK increase)
4 (8 mg/kg/d)	/	/	/

**Table 4 cancers-15-02020-t004:** Treatment-related adverse events were observed during all cycles with grade and dose levels (DL) at which they occurred.

Adverse Event	Grade 1	Grade 2	Grade 3	Grade 4
Fatigue	1 (DL1)	1 (DL2)		
Cough		1 (DL1)		
Nausea	2 (DL1, DL3)			
Constipation	1 (DL3)			
Diarrhea	1 (DL2)			
Abdominal pain	1 (DL3)	1 (DL2)		
Vomiting	2 (DL2, DL3)			
Oral mucositis	2 (DL2, DL3)			
Myalgia	1 (DL2)	1 (DL3)		
Maculopapular rash	1 (DL2)		1 (DL2)	
Blood CPK increase	1 (DL2)			1 (DL3)
Hyperkaliemia	1 (DL3)			
Transaminases increase			1 (DL3)	
Bilirubin increase			1 (DL3)	

**Table 5 cancers-15-02020-t005:** Summary of the fluvastatin and celecoxib pharmacokinetic parameters (mean ± standard deviation unless otherwise specified).

**Fluvastatin**									
Day	Dose	n	T_max_ (h) median (range)	C_max_ (ng/mL)	AUC_0–24 h_ (h.ng/mL)	AUC_0–∞_ (h.ng/mL)	T_1/2_ (h)	CL/F (L/h/kg)	V_z_/F (L/kg)
1 (cycle 1)	2 mg/kg	5	2 (1–5)	1238 ± 1030	2445 ± 1319	2459 ± 1324	4.6 ± 1.5	1.1 ± 0.7	6.5 ± 3.2
	4 mg/kg	7	2 (1–4)	4540 ± 5401	10,420 ± 10,189	10,460 ± 10,198	4.1 ± 1.3	1 ± 1	5.5 ± 6.1
	6 mg/kg	1	3	5336	14,515	14,535	3.7	0.4	2.2
14 (cycle 1)	2 mg/kg	5	2	1206 ± 1502	3375 ± 1988	-	4.8 ± 1	0.8 ± 0.6	5.9 ± 4.4
	4 mg/kg	4	1.5	6220 ± 7769	17,367 ± 20,413	-	4.8 ± 2	0.7 ± 0.5	2.9 ± 3.2
	6 mg/kg	1	5	4263	13,412	-	4.5	0.5	2.9
**Celecoxib**									
Day	Dose	n	T_max_ (h) median (range)	C_max_ (ng/mL)	AUC_0–12 h_ (h.ng/mL)	AUC_0–∞_ (h.ng/mL)	T_1/2_ (h)	CL/F (L/h/kg)	V_z_/F (L/kg)
1 (cycle 2)	200 mg	6	3.5 (2–4)	1475 ± 430	7577 ± 2032	11,065 ± 4222	6 ± 3	0.6 ± 0.5	4.2 ± 1.9
	400 mg	7	4 (3–8)	1351 ± 317	7378 ± 2228	10,696 ± 3631	6 ± 3	0.6 ± 0.2	5.0 ± 2.4
	200 mg	6	3.5 (2–4)	1475 ± 430	7577 ± 2032	11,065 ± 4222	6 ± 3	0.6 ± 0.5	4.2 ± 1.9

T_max_, time to maximum plasma concentration; C_max_, maximum plasma concentration; AUC_0–24 h_, area under the plasma concentration–time curve from 0 to 24 h post-dose; AUC_0–∞_, the area under plasma concentration-time curve extrapolated to infinity; T_1/2_, terminal elimination half-life; CL/F, apparent oral clearance normalized for body weight; and V_z_/F, apparent oral of volume of distribution during terminal phase normalized for body weight.

## Data Availability

Data supporting reported results can be obtained on request from the corresponding author.

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
