# Peer review of "Phase I Study of a Combination of Fluvastatin and Celecoxib in Children with Relapsing/Refractory Low-Grade or High-Grade Glioma (FLUVABREX)"

_cancers, 2023, doi:10.3390/cancers15072020_

Round 1
Reviewer 1 Report
Simple summary – The conclusions with regards to LGG are a bit overstated, I would decrease emphasis on future hypothesis generation and focus on the critical findings of this clinical trial.
I recommend clarifying the rationale for combining celecoxib and Fluvastatin in the introduction. While the authors mention Mercurio et al and the combination data in pilocytic astrocytoma, this gets lost a little in the subsequent detailed description of the individual compounds. Consider highlighting the proposed mechanism in a bit more detail and describing whether the anticipated toxicity profiles overlap.
Please include a CONSORT diagram or equivalent showing flow of enrolled subjects. Please clarify primary, secondary and exploratory endpoints (including PFS duration) in methods.
Dose-escalation design – the authors say they will evaluate 4 dose levels via CRML model for dose escalation. They then describe the probabilities of toxicity for doses of fulvastatin and state they will pick the dose level with the closest probability of DLT to 25%. However, they then select dose level 3, which seems further from a 25% probability of DLT than dose level 2. Statistical review and further explanation by the authors may be helpful here.
Expansion phase – in the methods the authors describe an expansion phase in which the number of subjects was increased to 14 evaluable patients at RP2D. However only 6 patients were evaluated (and reported) at dose level 3, which the authors choose as the RP2D. Please clarify whether the dose expansion cohort was recruited and is being reported on separately, or whether that expansion was not completed.
Discussion – the first paragraph (lines 63-66) appears to be a copy of the instructions that was not removed from the manuscript template.
Author Response
Please see tha attachment

Reviewer 2 Report
In the paper “Phase I study of a combination of fluvastatin and celecoxib in children with relapsing/refractory low-grade or high-grade glioma (FLUVABREX)” the authors report their experience on 20 patients.
Researching new or repurposed drugs, as in this case, whose combination might ameliorate PFS and reduce general toxicity is of paramount importance. The drugs investigated are particularly handy because of the oral administration. The main aim of the study was to determine the Recommended Phase 2 Dose (RP2D) of fluvostatin when combined with celecoxib and that was assessed after a dose-escalation phase to identify the dose-limiting toxicity (DLT). The expansion phase was limited to patients affected by low-grade gliomas and the treatment was administered beyond 1 year in case of a stable disease, after medical and patient agreement. The results showed very low toxicity with 1 patient developing cutaneous rush of grade 3 (with DLT level 2) and 1 patient with a grade 4 elevated CPK after DLT level 3. Then, the data obtained in this trial seems to assess the safety of Fluvabrex. Moreover, preliminary clues of efficacy have been suggested by the observation of stable disease in 8 patients out of 9 with low-grade glioma.
The study is well ideated and conducted, and a detailed description of the findings for each step has been provided. Then, a Phase II trial appears justified and desirable.
In the “Discussion” paragraph, the first sentence (the caption, rows 63-66) must be removed.
